# An Exploration of Seaweed Polysaccharides Stimulating Denitrifying Bacteria for Safer Nitrate Removal

**DOI:** 10.3390/molecules26113390

**Published:** 2021-06-03

**Authors:** Hui Zhang, Lin Song, Xiaolin Chen, Pengcheng Li

**Affiliations:** 1College of Marine Science and Biological Engineering, Qingdao University of Science and Technology, Qingdao 266042, China; zhanghui951227@163.com (H.Z.); 03231@qust.edu.cn (L.S.); 2CAS and Shandong Province Key Laboratory of Experimental Marine Biology, Center for Ocean Mega-Science, Institute of Oceanology, Chinese Academy of Sciences, Qingdao 266071, China; 3University of Chinese Academy of Sciences, Beijing 100049, China; 4Laboratory for Marine Drugs and Bioproducts, Pilot National Laboratory for Marine Science and Technology (Qingdao), No. 1 Wenhai Road, Qingdao 266237, China

**Keywords:** seaweed polysaccharides, denitrifying bacteria, nitrate removal, growth promotion

## Abstract

Excessive use of nitrogen fertilizer in intensively managed agriculture has resulted in abundant accumulation of nitrate in soil, which limits agriculture sustainability. How to reduce nitrate content is the key to alleviate secondary soil salinization. However, the microorganisms used in soil remediation cause some problems such as weak efficiency and short survival time. In this study, seaweed polysaccharides were used as stimulant to promote the rapid growth and safer nitrate removal of denitrifying bacteria. Firstly, the growth rate and NO_3_^−^-N removal capacity of three kinds of denitrifying bacteria, *Bacillus subtilis* (BS), *Pseudomonas stutzeri* (PS) and *Pseudomonas putida* (PP), were compared. The results showed that *Bacillus subtilis* (BS) had a faster growth rate and stronger nitrate removal ability. We then studied the effects of *Enteromorpha linza* polysaccharides (EP), carrageenan (CA), and sodium alginate (AL) on growth and denitrification performance of *Bacillus subtilis* (BS). The results showed that seaweed polysaccharides obviously promoted the growth of *Bacillus subtilis* (BS), and accelerated the reduction of NO_3_^−^-N. More importantly, the increased NH_4_^+^-N content could avoid excessive loss of nitrogen, and less NO_2_^−^-N accumulation could avoid toxic effects on plants. This new strategy of using denitrifying bacteria for safely remediating secondary soil salinization has a great significance.

## 1. Introduction

Rational use of nitrogen fertilizer can increase the yield of food crops, but when the use of nitrogen fertilizer exceeds the utilization capacity of plants, the crop yield cannot be further increased and introduce much nitrate. At the same time, due to the relatively closed environment of intensively managed agriculture, nitrate cannot be effectively discharged. Therefore, a large amount of nitrate is accumulated in the soil, which is the main cause of secondary soil salinization [1]. Reducing the nitrate is an effective way to repair the secondary soil salinization.

Denitrification refers to the process in which denitrifying microorganisms gradually reduce nitrate or nitrite to gaseous nitrogen (NO_3_^−^-NO_2_^−^-NO-N_2_O-N_2_) [2]. This process can convert nitrate into gas finally, thus effectively reducing nitrate content in soil. At present, denitrifying organisms are widely used in wastewater treatment [3,4,5]. However, few studies have used denitrifying bacteria for soil nitrate removal. The soil nitrate removal means the loss of nitrogen, but accumulated nitrate seriously affects the sustainable development of facilities agriculture. It is necessary to take effective measures to remove excessive nitrate in soil [6]. Appling denitrifying bacteria to soil for nitrate removal may be one of the effective ways for bioremediation of secondary salinized soil. *Bacillus* and *Pseudomonas* are the main biocontrol and plant growth-promoting bacteria [7,8], and it is feasible to select bacteria belonging to the genera of *Bacillus* and *Pseudomonas* as functional bacteria for secondary soil salinization bioremediation. Many research studies showed that BS, PS and PP have the denitrifying ability [9,10,11], so we compared the growth and nitrogen removal ability of these three bacteria, for the purpose of providing reference for selecting better bacteria to repair secondary soil salinization.

Seaweed is rich in polysaccharides such as alginate, carrageenan and fucoidan et al., which are different from other terrestrial plants. Seaweed polysaccharides are considered to be one of the most abundant organic molecules in the ocean. At present, the application of seaweed polysaccharides are mainly in the form of biological stimulant to promote seed germination, plant growth and induce plant defense response in agriculture [12,13,14,15]. However, few research studies have used the seaweed polysaccharides to remediate secondary soil salinization.

In order to select the most suitable bacteria for potential secondary soil salinization remediation, we compared the growth and denitrification capacity of BS, PS and PP. The growth rate of bacteria in liquid medium could reflect indirectly the survival ability of bacteria in salinized soil. For the safety of soil denitrification, the by-products of denitrification are also worth considering. Then, the effects of different seaweed polysaccharides on BS were studied. We evaluated the stimulating effect of seaweed polysaccharides on the growth of BS, and the addition of seaweed polysaccharides could reduce the accumulation of nitrite and the production of ammonia nitrogen, which provided a new idea for subsequent research on bioremediation of secondary soil salinization using denitrifying bacteria.

## 2. Results

### 2.1. Chemical Analysis of the Three Seaweed Polysaccharides

The sugar content of EP was measured, the molecular weight, total carbon content and monosaccharide composition of EP, AL and CA were measured, the sulfate content of EP and CA was also measured, as shown in Table 1.

Before the EP was extracted, the *Enteromorpha linza* was decolorized and defatted with 95% ethanol. The EP extracted was purified twice by dissolution and alcohol precipitation, so the total sugar content of the polysaccharide extracted from *Enteromorpha linza* was up to 96.50 ± 14.35%, more higher than 53.8% and 47.9% [16,17]. EP and CA belong to sulfate polysaccharides, the content of sulfate is 14.07%, 10.05%, respectively, and EP contains more sulfate groups. Most of the polysaccharides of green algae mainly consist of rhamnose, xylose and glucose, but less glucuronic acid [18]. The polysaccharide extracted from *Enteromorpha linza* in this study mainly contain rhamnose, glucuronic acid, xylose and glucose; the content of rhamnose is the highest, and the glucuronic acid is only second to rhamnose, which may be related to the season of algal harvesting, the method of sugar extraction [19]. In addition, the average molecular weights of EP, AL and CA were 756 kDa, 833 kDa and 1588 kDa, respectively. The average molecular weight of EP was the lowest and CA was the highest, which might affect their results for the growth of bacteria.

### 2.2. Comparison of the Growth and Denitrification Performance of Three Denitrifying Bacteria

#### 2.2.1. Comparison of the Growth of Three Denitrifying Bacteria

Figure 1 shows that BS had the fastest growth rate, followed by PP and PS, and the Lag Phase of BS was shorter, which indicated that BS had a stronger ability to adapt to the high nitrate environment.

In Figure 1, the growth of three bacteria (OD_600_) was well fitted with Slogistic 1 equation, y = a/(1 + exp(−*k* × (*t* − *t*_c_))). Just as shown in Table 2, the correlation coefficients (R^2^) of BS, PP and PS were 0.99, 0.97 and 0.99, respectively, indicating that the Slogistic 1 equation is applicable to fit the growth of these three bacteria.

The parameter ‘*a*’ indicated the maximum OD_600_ value. ‘*a*’ of BS, PP and PS were 0.82, 0.72 and 0.63, respectively (Table 2), indicating that BS showed the best growth in spite of a high concentration of nitrate culture. The parameter ‘*k*’ indicated the maximum growth rate. ‘*k*’ of BS, PP and PS (0.32, 0.30, 0.27) were slightly different and BS showed the best maximum growth rate, and parameter ‘*t_c_*’ indicated the time required to reach half of the maximum OD_600_ value. It also showed that BS took the shortest time to reach the maximum OD_600_ value.

#### 2.2.2. The Changes of Inorganic Nitrogen Concentration in Fermentation Liquid of Bacillus Subtilis (BS), Pseudomonas Stutzeri (PS) and Pseudomonas Putida (PP)

Figure 2a shows that all the three bacteria could use NO_3_^−^-N as a nitrogen source. It can be seen that the three bacteria consumed most of the NO_3_^−^-N within 24 h. In addition, the NO_3_^−^-N concentration changed slightly after 24 h, which was consistent with the growth curve of the three bacteria, and the consumption of NO_3_^−^-N during the exponential growth period was higher than the stable growth period. BS and PP had similar NO_3_^−^-N consumption capacity before 24 h, but BS performed better after 24 h. At 36 h, BS, PP and PS decreased by 18.2%, 14.0% and 15.8%, respectively, indicating that BS has a higher ability to consume much nitrate.

The NO_2_^−^-N was accumulated shortly after the culture process of the three kinds of bacteria (Figure 2b). It showed that the NO_2_^−^-N concentration of BS and PP reached the maximum at 6 h, and it was 4.68 ± 0.31 and 0.83 ± 0.09 mg/L, respectively. However, the NO_2_^−^-N concentration of the PS reached maximum value 1.25 ± 0.10 mg/L at 12 h, which was six hours later than that of BS and PP. This was consistent with the growth and nitrate curves. There was no obvious accumulation of NO_2_^−^-N in the three bacteria after 18 h. This might be a result of increased NO_2_^−^-N invertase after 6 h of BS and PP and after 12 h of PS.

As shown in Figure 2c, the NH_4_^+^ content of BS increased gradually during the culture process, while that of PS and PP increased slightly. BS reached maximum value of 4.99 ± 0.63 mg/L at 36 h, while PS and PP had no obvious accumulation. The result indicated that BS might have a stronger ability of dissimilatory nitrate reduction to ammonium (DNRA) than PP and PS.

It was obvious that the TN-N content of all three bacteria decreased during the culture process (Figure 2d), while that of BS was the lowest before 18 h, which might be due to the fast growth rate and high biomass of BS; more nitrogen sources were needed to sustain its own growth and metabolism. BS showed an increase in TN-N content at 24 h, which might be related to exocrine and the release of intracellular substances caused by dead cell. The trend of total nitrogen reduction of the three bacteria was similar, and there was no significant difference after 30 h.

In conclusion, BS presented the best growth and nitrate removal ability in a high-concentration nitrate environment, so we chose BS to carry out the follow-up experiments.

### 2.3. Effects of Seaweed Polysaccharides on the Growth and Denitrification of Bacillus subtilis (BS)

#### 2.3.1. Effects of Seaweed Polysaccharides on the Growth of *Bacillus subtilis* (BS)

In Figure 3, the OD_600_ of all treatment groups were higher than that of the control group during 6–30 h. The OD_600_ of AL, EP, CA and CK reached maximum values at 24 h, and were 0.9990 ± 0.0176, 0.9853 ± 0.0212, 0.9607 ± 0.0231, 0.9289 ± 0.0247, respectively. The significant difference of biomass between treatment groups and the control group suggested that the addition of seaweed polysaccharides effectively promoted the growth of BS. The biomass of the AL and EP groups was higher than that of the CA group at 24 h, which might be related to the different characteristics of these three seaweed polysaccharides.

#### 2.3.2. The Changes of Inorganic Nitrogen Concentration in Fermentation Liquid of *Enteromorpha linza* Polysaccharides (EP), Carrageenan (CA), Sodium Alginate (AL) and Control (CK) Groups

As we can see in Figure 4, there was no significant difference in NO_3_^−^-N consumption between the treatments and the control group before 6 h, but there appeared an obvious difference at 12 h. The concentrations of NO_3_^−^-N in EP, CA, AL and CK were 181.63 ± 0.90, 186.83 ± 1.50, 247.24 ± 11.51 and 249.04 ± 21.59 mg/L at 12 h and the reduction rates (RrNO_3_^−^ = (initial concentration-final concentration)/initial concentration) were 61.11%, 60.12%, 46.98% and 45.37%, respectively. However, the concentration of NO_3_^−^-N in EP and CA was lower than that of AL and CK at 18 h, although the growth of BS for AL at 18 h was higher than that of CA. There was no significant difference between the treatments and the control group after 24 h. The results indicated that the addition of seaweed polysaccharides could improve the NO_3_^−^-N consumption rate of BS. During 18 h of culture, the effect of EP was the best among the three polysaccharides.

Figure 5 shows that NO_2_^−^-N content in EP, CA, AL and CK reached the maximum accumulation at 12 h, with concentrations of 52.80 ± 3.48, 72.15 ± 4.53, 40.50 ± 7.26 and 84.48 ± 8.56 mg/L, respectively. The results showed that the addition of seaweed polysaccharide alleviated the accumulation of NO_2_^−^-N, although Figure 4 indicates that the consumption of NO_3_^−^-N in the EP, CA and AL groups was higher than that in CK, which may indicate that the addition of seaweed polysaccharides improved the ability of BS to produce gaseous nitrogen or dissimilated reduction. After 12 h, the concentration of NO_2_^−^-N in the seaweed polysaccharides groups and the control group began to decrease, and after 24 h, there was no significant difference in the content of NO_2_^−^-N in each system.

As shown in Figure 6, there was no significant difference in NH_4_^+^-N content between the seaweed polysaccharides groups and the control group before 6 h, but the accumulation of NH_4_^+^-N significantly changed at 12 h. At this time, the NH_4_^+^-N content levels of EP, AL, CA and CK were 176.96 ± 4.95, 175.33 ± 9.46, 162.03 ± 0.73 and 154.80 ± 9.98 mg/L, respectively. At 18 h, the accumulation of NH_4_^+^-N in EP and AL reached the maximum, and the contents were 307.50 ± 0.13 and 305.12 ± 5.47 mg/L, respectively. NH_4_^+^-N accumulation of CA and CK reached the maximum at 24 h. From 12 to 24 h, NH_4_^+^-N accumulations of the EP, AL and CA groups were higher than that of CK. It might suggest that the addition of seaweed polysaccharides could promote the DNRA process, which partly explained that the seaweed polysaccharides could reduce the NO_2_^−^-N accumulation. From the whole culture stage, there was no significant difference in the maximum accumulation of NH_4_^+^-N in the EP, AL and CA treatment groups, but the EP and AL treatment groups could reach the maximum accumulation of NH_4_^+^-N faster. There was a small decrease in NH_4_^+^-N in the treatment groups and the control group in the later period, which might be related to the small consumption of NH_4_^+^-N by BS [20] or the release of NH_3_.

The TN-N were decreased continuously in all treatment groups and the control group before 18 h (Figure 7), while the TN-N of all the groups reached their minimum value at 18 h except for the CA group, and the TN-N content began to increase after 18 h; this might be related to the release of intracellular substances of dead cell. At 12 h, the TN-N content of the CK group was higher than the treatment groups, which might indicate that the addition of seaweed polysaccharides promoted the secretion of extracellular substances, such as enzyme proteins and polysaccharides, etc. The TN-N of the EP treatment group reached the lowest content at 18 h, followed by CK, AL and CA, whose concentrations were 363.28 ± 2.54, 374.07 ± 13.68, 375.34 ± 5.29 and 402.83 ± 1.90 mg/L, respectively. This might be related to the fact that EP stimulated BS to remove nitrogen faster and more.

## 3. Discussion

Owing to the high rate of continuous cropping, high use of nitrogen fertilizer beyond the need of vegetables and rarely rainwater leaching, the salt accumulated gradually in intensively managed agriculture. The excessive nitrate was the typical characteristic of secondary soil salinization, which caused serious impacts on the yield and quality of vegetables and restricted the sustainable development of agriculture. Therefore, how to reduce the nitrate content was the key to the repair of secondary soil salinization [21,22].

At present, adding the biological agents and organic fertilizer, compound microbial fertilizer is the common method to remediate the salinized soil. Biological remediate secondary soil salinization was an eco-friendly and sustainable method for agriculture. Microorganisms remediate secondary salinization was one of the promising methods of biological improvement measures. However, there are some problems such as weak pertinence of microorganisms used in bioremediation and unclear effects of metabolites on soil. At the same time, many factors could affect the activity of functional microorganisms, such as soil salinity, pH, humidity, temperature, etc. [23], leading to short survival time and slow growth of functional microorganisms and long time to repair. Therefore, promoting the growth of functional microorganisms might be an effective way to remediate the secondary salinized soil.

This study firstly compared the growth and NO_3_^−^-N removal ability of BS, PP and PS in DM. The results indicated that BS had a higher growth rate than PP and PS, and the NO_3_^−^-N consumption was agreeable with the growth curve. After 18 h, the concentration of NO_3_^−^-N in the BS group was significantly lower than that in the other two groups, which indicated that the NO_3_^−^-N consumption capacity of BS was the best.

Figure 3 shows that the three denitrifying bacteria all accumulated NO_2_^−^-N at the beginning 6 h of the culture process. Although the accumulated NO_2_^−^-N content of the BS group was higher, there was no significant difference in the NO_2_^−^-N content of the three treatment groups after 18 h. The result was not in accordance with that of one reference [24], which inferred that BS could accumulate nitrite in the culture process. This might be due to differences of species. While the accumulation of NO_2_^−^-N of the PS group was similar to the literature [25] that also showed the phenomenon of increasing at first and then decreasing. Some references reported that no accumulation of NO_2_^−^-N was found in the PP culture process [26,27]. However, a transient accumulation was found in this study, which might be related to the differences of bacteria strains and culture conditions.

NH_4_^+^-N content of the BS group increased gradually, similar to the literature [21] that reported that BS has a strong dissimilatory reduction ability; nevertheless, *Pseudomonas* groups (PP and PS) had no obvious ammonia nitrogen accumulation. This might result in the excessive loss of nitrogen sources in soil if the functional bacteria only have a strong denitrification ability.

Figure 2d showed that BS had the fastest decline in TN-N, and this might be related to the higher growth and NO_3_^−^-N reduction rate, indicating that BS has strong adaptability to the environment with high nitrate content, and this provided the possibility for the remediation of secondary salinized soil, which has complex interference factors.

After comparing three kinds of denitrifying bacteria growth and nitrogen removal ability, we studied the effects of seaweed polysaccharides on the growth and the nitrogen removal of BS. The results showed that the addition of seaweed polysaccharides promoted the growth of BS, which might be because the seaweed polysaccharides increased the carbon source in the culture medium. However, the results did not display diauxie, indicating that BS could consume the glucose and seaweed polysaccharides simultaneously [28]. Although the total carbon of these three seaweed polysaccharides added into the DM was the same, there were some differences in their growth performance and the nitrogen transformation, which might relate to the different structure, sulfate content, monosaccharide composition and molecular weight of these seaweed polysaccharides [29,30], but further research was needed.

The reducing trends of NO_3_^−^-N matched the growth curve, and the NO_3_^−^-N content in the EP, CA groups had obvious differences compared to the CK group at 12 h; the addition of seaweed polysaccharides promoted the NO_3_^−^-N consumption, which might possess the similar effect when using the seaweed polysaccharides in soil. There was no obvious difference between the treatment and CK groups after 18 h, which was because seaweed polysaccharides were added to the DM in the form of additive, while glucose can provide adequate carbon source for cell growth.

Appeared accumulation of NO_2_^−^-N should be paid attention to regarding the denitrification process in soil, because the NO_2_^−^-N is toxic to plants, and restrains the growth of plants [31,32], and compared to the control group, the adding of seaweed polysaccharides could reduce the accumulation of NO_2_^−^-N. It can be seen from Figure 4 that the addition of seaweed polysaccharides can promote the ability of dissimilatory reduction of BS, which partly explains the reason for the less accumulation of NO_2_^−^-N in seaweed polysaccharides groups.

At the same time, the formation of NH_4_^+^-N can avoid the excessive loss of nitrogen in the form of gas such as N_2_, NO and N_2_O in the process of denitrification [33,34], and it could also balance ammonia: nitrate ratio that was beneficial for plant growth [35].

The decreases of TN-N content in the seaweed polysaccharides groups were slower. This might be because seaweed polysaccharide promoted the cell body to secrete some substances including N, such as proteins, polysaccharides and other substances. The increase of TN-N content after 18 h might be due to the release of substances contained in dead cell.

## 4. Materials and Methods

### 4.1. Materials and Bacteria

*Bacillus subtilis* (BS), *Pseudomonas stutzeri* (PS) and *Pseudomonas putida* (PP) were purchased from BeNa Culture Collection. These bacteria were further purified and identified. Then, bacteria strains were stored at 4 ℃ for subsequent experiments. *Enteromorpha linza* was collected in the Second Bathing Beach of Qingdao on July 2019, and the collected *Enteromorpha linza* was cleaned and dried in the laboratory. After that, the dried *Enteromorpha linza* was crushed into powder and stored in a plastic bag at room temperature.

Composition of denitrification medium (DM, 1.3L): KNO_3_ 4.68 g, KCl 1.3 g, KH_2_PO_4_ 0.65 g, FeSO_4_.7H_2_O 13 mg, MgSO_4_.7H_2_O 0.65 g, CaCl_2_ 1.3 mg, glucose 16.25 g, pH = 7.0.

### 4.2. Preparation of Seaweed Polysaccharides

The extraction method [36] of seaweed polysaccharide was modified slightly: 100 g of dried *Entermorpha linza* powder were put into a three-necked flask with 1 L 95% ethanol. The powder was defatted and decolorized after reflux at 80 ℃ for 2 h and repeated again. Then, the treated *Entermorpha linza* was dried in an oven at 70 ℃ and mixed with 3 L distilled water, and the mixture was treated in a sterilizer at 100 ℃ for 4 h. After that, the supernatant was obtained by centrifugation at 10,000 rpm. It was concentrated in vacuum. Then, four times volume of 95% ethanol was added to the supernatant and precipitated overnight at 4 ℃. Then the mixture was centrifuged to obtain polysaccharide precipitation which was dissolved in distilled water and dialyzed by a 3500 Da dialysis bag for two days. At last, the polysaccharides were freeze-dried and stored in the plastic bag.

The carrageenan (Tianjin aopusheng Chemical Co. Ltd, Tianjin, China, analytical grade) and sodium alginate (Sinopharm Chemical Reagent Co. Ltd, Shanghai, China, analytical grade) samples were dissolved in an appropriate amount of hot distilled water at 100 ℃. It was also dialyzed by a 3500 Da dialysis bag for two days. Finally, the samples were freeze-dried and stored in the plastic bag.

### 4.3. Chemical Analysis of Seaweed Polysaccharides

The Mw of seaweed polysaccharides were determined by high-performance gel permeation chromatography (HPGPC) with 0.05 M Na_2_SO_4_ aqueous solution as mobile phase on an Agilent 1260 HPLC system (Agilent Technologies, Inc., Santa Clara, CA, USA). Dextran standards with Mw 13, 64, 135, 300, 670 kDa (Sigma-Aldrich, Inc., St. Louis, MO, USA) were used to calibrate the column. Total sugar was analyzed using the phenol–sulfuric acid method [37]. Total carbon and sulfur contents were determined by elemental analysis. The molar ratio of monosaccharide composition was analyzed by 1-phenyl-3-methyl-5-pyrazolone (PMP) precolumn derivation HPLC [38]. The solution of samples (10 mg/mL) were hydrolyzed in 4 M trifluoroacetic acid in a 10 mL ampoule for 4 h at 110 °C. Then, the hydrolyzed mixture was derivatized and neutralized to pH = 7.0. Later, the mixture was extracted by trichloromethane and filtered (0.22 μm filter), then the aqueous phase was loaded onto a ZORBAX SB-AQ column and analyzed by Shimadzu SPD-M20A high-performance liquid chromatography (HPLC). Elution was conducted with phosphate buffer (pH = 6.7): Acetonitrile (83:17) for 45 min with UV detection (DAD 254 nm).

### 4.4. Comparison of the Growth and Denitrification Capacity of Three Denitrifying Bacteria

BS, PS and PP were cultured overnight in LB medium and Beef Extract-Peptone medium, respectively. Then, the three bacteria were inoculated with the same original OD_600_. DM without inoculation was used as control. All the groups were cultured at 30 ℃, 140 rpm in incubator. Next, 3 mL of samples were taken every 6 h, and we measured OD_600_ with an enzyme plate analyzer (Infinite M1000Pro). Then, every sample was centrifuged, and the supernatant was stored at −80 ℃ for subsequent concentration determination of NO_3_^−^-N, NO_2_^−^-N, NH_4_^+^-N, TN-N. All samples were analyzed following the standard methods [39].

Method for NO_3_^−^-N measure: different volumes (0, 2, 4, 6, 8, 10 mL) of NO_3_^−^-N standard solution (10 μg/mL) were added into a 25 mL colorimetric tube, and then we added deionized water to the marking line. After shaking, the absorbance at 220, 275 nm was measured with a colorimetric dish (optical path 10 mm). The relationship between nitrate concentration and absorbance is plotted according to A_NO3^−^_ = A_220_ − 2A_275_. The samples were diluted, and 1 mL 1% sulfonic acid solution was added, respectively. After shaking, the absorbance at A_220_ and A_275_ was determined. The concentration of NO_3_^−^-N was calculated according to the formula above.

Method for NO_2_^−^-N measure: different volumes (0, 1, 3, 5, 7, 10 mL) of 1 μg/mL standard solution of NO_2_^−^-N were diluted to the 10 mL marking line with deionized water. We added 1 mL chromogenic agent (including 2 g/L N-(1-naphthalyl)-ethylenediamine dihydrochloride, 40 g/L sulfanilamide, 8.5% phosphoric acid) and mixed well. The diluted sample underwent the same treatment and stood for 20 min. The absorbance was measured within 2 h using enzyme plate analyzer at 540 nm, and then we calculated the amount of NO_2_^−^-N according to the standard curve.

Method for NH_4_^+^-N measure: different volumes (0, 1, 2, 4, 6, 8 mL) of 1 μg/mL ammonium standard solution were added into the colorimetric tube and diluted to about 8 mL with deionized water. Then, 1 mL of colorimetric agent (including 50 g/L of salicylic acid, 50 g/L of potassium sodium tartrate, pH = 6.0–6.5) and 2 drops of sodium nitroferricyanide solution (1%) were added and mixed well. Then, 2 drops of sodium hypochlorite solution (effective chlorine: 0.35%, free alkali: 0.75 mol/L) were added and diluted to the line with deionized water and mixed well. The diluted sample underwent the same treatment and stood for 1 h and was then measured at 697 nm with a microplate analyzer. Then calculated the amount of NH_4_^+^-N according to the standard curve.

Method for TN-N measure: different volumes of (0, 0.5, 1, 2, 3, 5, 7, 8 mL) 10 μg/L potassium nitrate standard solution were diluted with deionized water to the 10 mL marking line in a 25 mL colorimetric tube. We added 5 mL of alkaline potassium persulfate solution, plugged the grinding plug, and wrapped the tube plug with gauze and yarn rope to prevent it from spilling out. We put the colorimetric tube in the sterilization pot at 126 ℃ for 50 min. After the sterilization pot was finished, we took out the colorimetric tube and cooled it naturally to room temperature when the pressure pointer was zero again. We added 1 mL of hydrochloric acid (1 + 9) and diluted it with deionized water to the 25 mL marking line. The diluted samples were similarly treated and measured on an ultraviolet spectrophotometer. The absorbance was measured at 220 nm and 275 nm using a colorimetric dish (optical path 10 mm), The relationship between nitrate concentration and absorbance is plotted according to A_NO3_- = A_220_ − 2A_275_, the concentration of TN-N (NO_3_^−^-N) was calculated according to this formula.

### 4.5. Effects of Seaweed Polysaccharides on the Growth and Denitrification Capacity of Bacillus subtilis (BS)

Different seaweed polysaccharide solutions which have the same carbon content were sterilized at 121 ℃ for 20 min. After cooling, 3 mL of the EP solution (EP group), carrageenan solution (CA group), sodium alginate solution (AL group) and sterile water (CK group) were, respectively, added into conical flasks with 1% (*v*/*v*) bacteria suspension and cultured. There were three replicates for each group. Every 6 h, 3 mL of fermentation liquid was taken out from each conical flask. Then, 200 μL of each sample were taken into 96-well plates to measure OD_600_ with enzyme plate analyzer (Infinite M1000Pro). The rest of the fermentation liquid was centrifuged, and the supernatant was taken into centrifuge tubes and stored at −80 ℃ for subsequent measurement of NO_3_^−^-N, NO_2_^−^-N, NH_4_^+^-N and TN-N.

### 4.6. Statistical Analysis

The results are presented as the average (mean value) and standard deviation of three experiments with duplicate. Statistical analysis was performed using ORIGIN9.1. Differences between means were considered statistically significant if *p* < 0.05.

## 5. Conclusions

This study firstly compared the growth and nitrogen removal of the three kinds of denitrifying bacteria, and found that *Bacillus subtilis* had a faster growth rate, stronger NO_3_^−^-N consumption rate and greater ability of dissimilatory nitrate reduction to ammonium than *Pseudomonas stutzeri* (PS) and *Pseudomonas putida* (PP). Therefore, we chose *Bacillus subtilis* as the object of subsequent research. We found that the addition of seaweed polysaccharides could significantly promote the growth of *Bacillus subtilis*, accelerate the NO_3_^−^-N consumption, decrease the accumulation of NO_2_^−^-N, and promote the accumulation of NH_4_^+^-N. This study provided a new idea to remediate the secondary soil salinization, but further studies are necessary in the future.

## Figures and Tables

**Figure 1 molecules-26-03390-f001:**
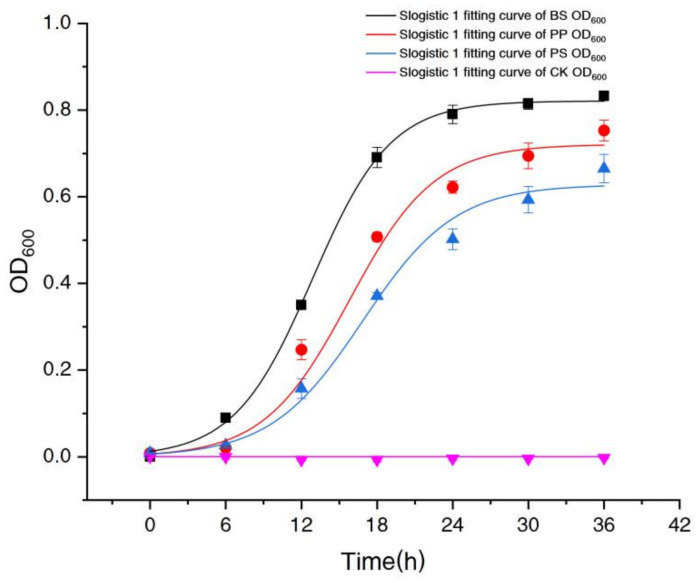
Fitting growth curves of *Bacillus subtilis* (BS), *Pseudomonas stutzeri* (PS) and *Pseudomonas putida* (PP).

**Figure 2 molecules-26-03390-f002:**
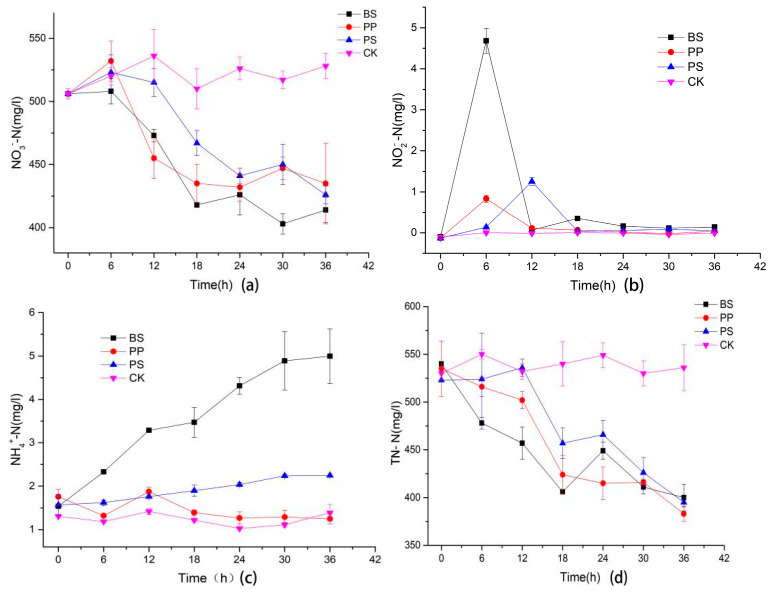
The concentration changes of NO_3_^−^-N, NO_2_^−^-N, NH_4_^+^-N, TN-N, (**a**) represents the NO_3_^−^-N, (**b**) represents the NO_2_^−^-N, (**c**) represents the NH_4_^+^-N, (**d**) represents the TN-N.

**Figure 3 molecules-26-03390-f003:**
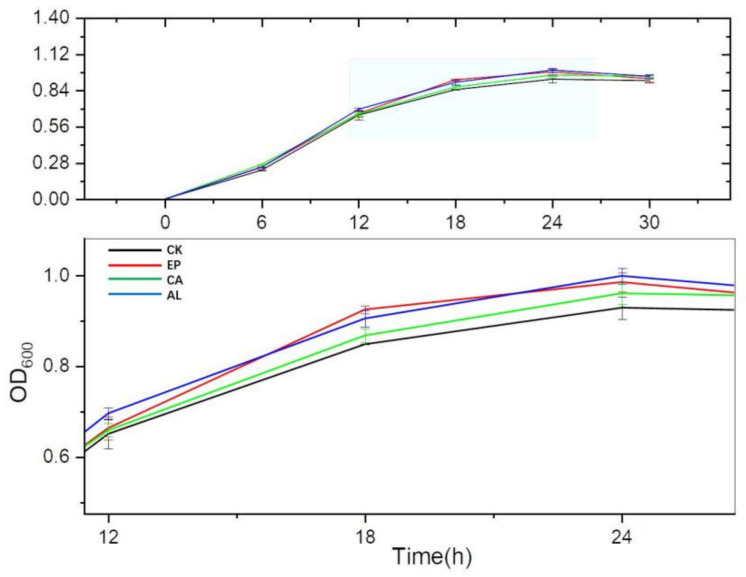
Effects of *Enteromorpha linza* polysaccharides (EP), carrageenan (CA), sodium alginate (AL) on the growth of *Bacillus subtilis* (BS).

**Figure 4 molecules-26-03390-f004:**
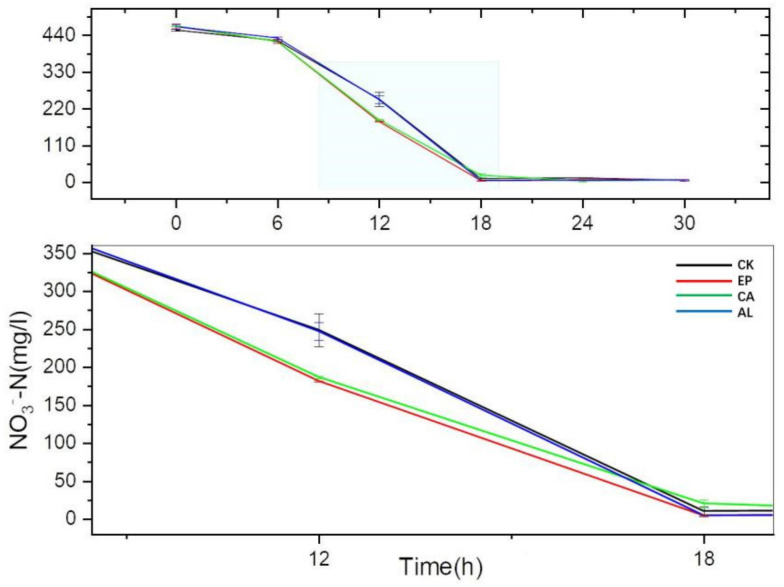
The changes of NO_3_^−^-N concentration with culture time.

**Figure 5 molecules-26-03390-f005:**
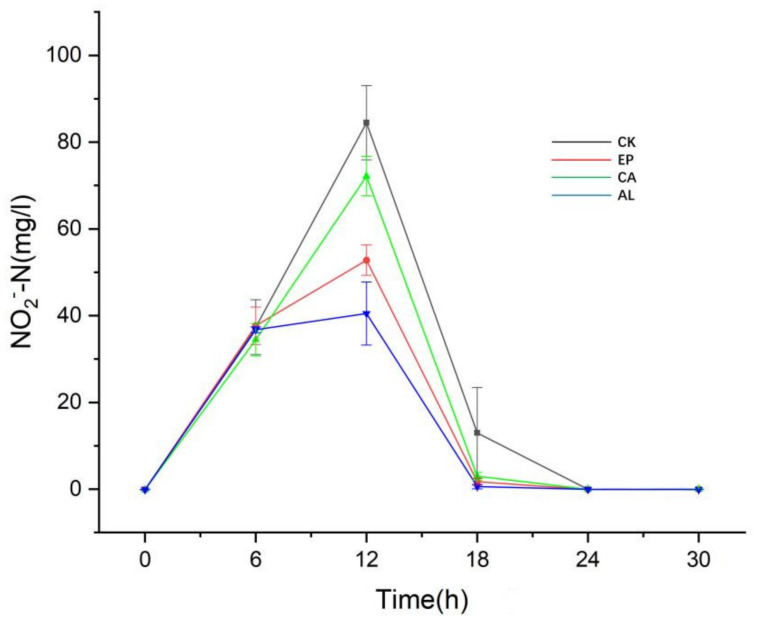
The changes of NO_2_^−^-N concentration with culture time.

**Figure 6 molecules-26-03390-f006:**
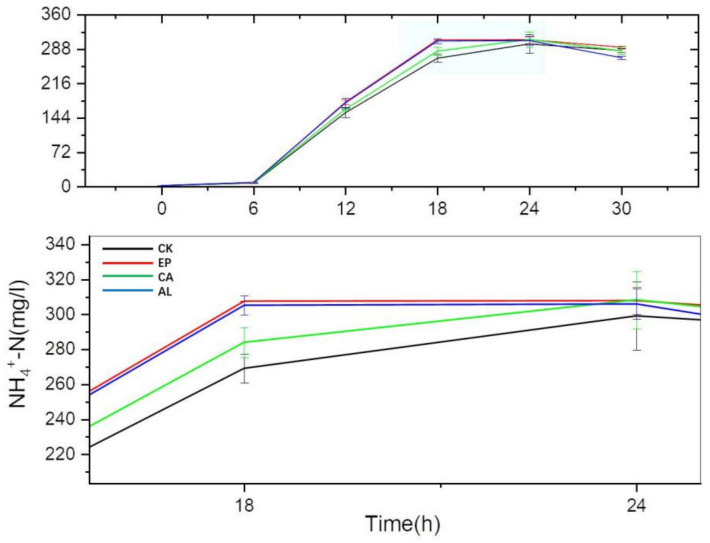
The changes of NH_4_^+^-N concentration with culture time.

**Figure 7 molecules-26-03390-f007:**
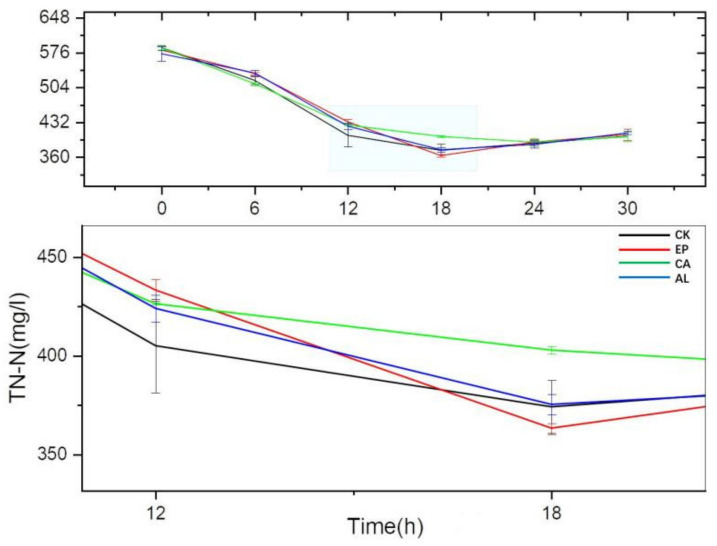
The changes of TN-N concentration with culture time.

**Table 1 molecules-26-03390-t001:** Chemical analysis and composition.

Sample	Total Sugar (%)	Sulfate (%)	Mw (kDa)	TC (%)	Composition of Monosaccharide (in Molar Ratio)								
					Rha	GlcUA	Xyl	Glc	Gal	Man	Fuc	GulA	ManA
**EP**	96.50 ± 1.31	14.07	756	31.23%	1.00	0.49	0.26	0.08	0.04	0.02	0.02	-	-
**CA**	-	10.05	1588	34.06%	-	-	-	0.61	0.61	1.00	-	-	-
**AL**	-	-	833	27.16%	0.41	-	-	0.51	-	-	-	0.15	1.00

Mw: molecular mass; TC: total carbon; Rha: rhamnose; GlcUA: glucuronic acid; Xyl: xylose; Glc: glucose; Gal: galactose; Man: mannose; Fuc: fucose; GulA: guluronic acid; ManA: mannuronic acid.

**Table 2 molecules-26-03390-t002:** Comparison of fitting parameters of *Bacillus subtilis* (BS), *Pseudomonas stutzeri* (PS) and *Pseudomonas putida* (PP).

	BS	PP	PS
*a*	0.82	0.72	0.63
*t_c_*	12.88	15.78	16.89
*k*	0.32	0.30	0.27
R^2^	0.99	0.97	0.99

## Data Availability

Not applicable.

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
