# Peer review of "An Exploration of Seaweed Polysaccharides Stimulating Denitrifying Bacteria for Safer Nitrate Removal"

_molecules, 2021, doi:10.3390/molecules26113390_

Round 1

Reviewer 1 Report

The article "An exploration of seaweed polysaccharides stimulating denitrifying bacteria for safer nitrate removal" deals with a natural way to address secondary soil salinization and address the accumulation of the nitrates in the soil. The authors have developed a new strategy of using denitrifying bacteria for safely remediate secondary soil salinization. The results provide an advance in current knowledge on the use of marine polysaccharides as carbohydrate source for the denitrifying bacteria. Based on the growth rate, NO3--N consumption  and the ability of dissimilatory reduction, Bacillus subtilis was selected as the most suitable bacteria for potential secondary soil salinization remediation. The article is written in an appropriate way, with some minor issues in presentation. The study is correctly designed and conducted. The methods for the concentration determination of NO3--N, NO2--N, NH4+-N, TN-N should be described with sufficient details to allow another researcher to reproduce the results.

Please put all the binomial names in italic (e.g. l 22, 78, 79 etc.)

The authors may introduce the abbreviation for Enteromorpha linza polysaccharide and may use only the abbreviation afterwards.

All Table titles and Figure captions: please put the meaning of all the abbreviations.

l. 114 "2.2.2. The Changes of NO3--N, NO2--N, NH4+-N, TN-N in Fermentation Liquid of BB, PP, PS" Please re-write the title with minimum use of abbreviations.

"group" to "groups" line 85

l 357 "other two pseudomonas" plese writhe the names of the organisms

Please put a space sign between the number and the unit throughout the manuscript.

Author Response

Responses to reviewers:

Reviewer 1:

  1. The methods for the concentration determination of NO3--N, NO2--N, NH4+-N, TN-N should be described with sufficient details to allow another researcher to reproduce the results.

Response: We have rewritten the detailed method according to your comment.It mainly included the drawing of the standard curve, the concentration of the reagent used, the detection wavelength and the instrument used.

Please put all the binomial names in italic (e.g. l 22, 78, 79 etc.)

Response: We have italicized  “Enteromorpha Linza” to “Enteromorpha Linza”

The authors may introduce the abbreviation for Enteromorpha linza polysaccharide and may use only the abbreviation afterwards.

Response: We have changed from “Enteromorpha Linza polysaccharide to “EP”.

All Table titles and Figure captions: please put the meaning of all the abbreviations.

Response: We have explained the meaning of abbreviations of all the table titles and Figure captions.

  1. 114 "2.2.2. The Changes of NO3--N, NO2--N, NH4+-N, TN-N in Fermentation Liquid of BB, PP, PS" Please re-write the title with minimum use of abbreviations.

Response: we have rewritten the titles, for example “2.2.2. The Changes of inorganic nitrogen concentration in Fermentation Liquid of Bacillus subtilis (BS), Pseudomonas stutzeri (PS) and Pseudomonas putida (PP); 2.3.2. The Changes of inorganic nitrogen concentration in Fermentation Liquid of E. linza polysaccharides (EP), carrageenan (CA), sodium alginate (AL) and control (CK) Groups”.

"group" to "groups" line 85

Response: We have revised "group" to "groups"

l 357 "other two pseudomonas" please write the names of the organisms

Response: We have revised "other two Pseudomonas" to "Pseudomonas stutzeri (PS) and Pseudomonas putida (PP)"

Please put a space sign between the number and the unit throughout the manuscript.

Response: we have put a space sign between the number and the unit throughout the manuscript, for example,”4.68g” to “4.68 g”, “4℃” to “4 ℃”.

Reviewer 2 Report

for starters I would like to congratulate the team for the article, which presents a very interesting study and at the same time very important for the field of environmental engineering.

However, a number of changes must be made to the paper (these changes do not have the role of diminishing the value of the work):

  • the paper does not present the research methodology and how this research was conducted
  • the paper touches on a very important field, the removal of nitrates from the soil, but my questions are (and I think should be put in the paper): the type of soil under analysis (geological structure), the soil analyzed how long it was used in agriculture, what crops have been (say in the last 5 years), from what depth was take the sample of the soil, the sample was taken destructively or non-destructively (natural sample)
  • now about the figures:

- please put figures 1, 2 and 4 again (clear figures)

- figures 2 and 4 can be made larger (they are too small and i do not understood)

- from the analysis of the figures, these are made with the origin program, but I think that some of the figures made by you can be made much more clearly using another type of representation from the origin - for figures 3 and 4. please look at this link - https://www.originlab.com/doc/Origin-Help/Zoom-Graph

  • in the paper, in the text, you also have an equation (line 103) y=a/(1+exp(-k*(x-xc))). mathematically it is not written correctly.  Correct is y=a/(1+exp^(-k*(x-xc))). and please pay close attention to the terms of the equation (these are presented in table 2), more precisely at the term Xc (Xc in table and xc in equation and text).   I don't know what x represents
  • on line 103 you have R2. But in table 2 you have R^2 and this is correct
  • you made a series of BS, PP and PS notations as an example. the group of words that I want to write in this way, at the first appearance in paper i put this notation,  not in chapter 4, change and put in chapter 1 line 50-51

Author Response

Responses to reviewers:

Reviewer 2:

However, a number of changes must be made to the paper (these changes do not have the role of diminishing the value of the work):

  • the paper does not present the research methodology and how this research was conducted

Response: We have revised the introduction according to your comment. We described the purpose of the study and the methods used to assess what types of denitrifying bacteria are suitable for safer soil denitrification.

  • the paper touches on a very important field, the removal of nitrates from the soil, but my questions are (and I think should be put in the paper): the type of soil under analysis (geological structure), the soil analyzed how long it was used in agriculture, what crops have been (say in the last 5 years), from what depth was take the sample of the soil, the sample was taken destructively or non-destructively (natural sample)

Response: Your opinion is very important, our goal is to alleviate the secondary salinization in greenhouse culture environment. The degree of salinization is related to planting time, crop type and soil texture, but because of the complexity of the soil,  a lot of studies need to involve in soil experiments, such as whether the denitrifying bacteria can quickly multiply in salinity soil, the effects of bioenhancement on soil microbial community, soil structure, soil composition and plant growth all need to be considered. The research of soil needs more detailed discussion, which is also our next work. Although this study does not directly involve soil environment, we have seen a good trend, and we can also see its great potential in soil remediation.

  • now about the figures:

- please put figures 1, 2 and 4 again (clear figures)

- figures 2 and 4 can be made larger (they are too small and i do not understood)

- from the analysis of the figures, these are made with the origin program, but I think that some of the figures made by you can be made much more clearly using another type of representation from the origin - for figures 3 and 4. please look at this link - https://www.originlab.com/doc/Origin-Help/Zoom-Graph

Response: We have put clear figures 1, 2 and 4 again, and lager figures 2 and 4. Figures 3 and 4 have been revised according to your comment.

  • in the paper, in the text, you also have an equation (line 103) y=a/(1+exp(-k*(x-xc))). mathematically it is not written correctly.  Correct is y=a/(1+exp^(-k*(x-xc))). and please pay close attention to the terms of the equation (these are presented in table 2), more precisely at the term Xc (Xc in table and xc in equation and text).   I don't know what x represents

Response: We have made some changes, The equation”y=a/(1+exp(-k*(t-tc)))” is based on a reference(”Zhao, B.; Cheng, D.Y.; Tan, P.; An, Q.; Guo, J.S. Characterization of an aerobic denitrifier Pseudomonas stutzeri strain XL-2 to achieve efficient nitrate removal. Bioresour. Technol. 2018, 250, 564–573. https://doi.org/10.1016/j.biortech.2017.11.038), We have revised the “xc” and “x”to “tc” in equation and text.

  • on line 103 you have R2. But in table 2 you have R^2 and this is correct

Response: We have revised the”R2” to “R^2”.

  • you made a series of BS, PP and PS notations as an example. the group of words that I want to write in this way, at the first appearance in paper i put this notation,  not in chapter 4, change and put in chapter 1 line 50-51

Response: we have changed and put the “BS, PP and PS notations” in paper when it first appeared.